# Reconsidering the Tolerable Upper Levels of Zinc Intake among Infants and Young Children: A Systematic Review of the Available Evidence

**DOI:** 10.3390/nu14091938

**Published:** 2022-05-05

**Authors:** Sara Wuehler, Daniel Lopez de Romaña, Demewoz Haile, Christine M. McDonald, Kenneth H. Brown

**Affiliations:** 1Nutrition International, Global Technical Services, Ottawa, ON K2P 2K3, Canada; dlopezderomana@nutritionintl.org; 2Department of Nutrition, Institute for Global Nutrition, University of California, Davis, CA 95616, USA; dewolde@ucdavis.edu; 3Institute for Health Metrics and Evaluation, University of Washington, Seattle, WA 98195, USA; 4Department of Pediatrics, University of California, San Francisco, CA 92161, USA; christine.mcdonald@ucsf.edu; 5International Zinc Nutrition Consultative Group, Oakland, CA 94609, USA

**Keywords:** review, zinc, upper-intake levels, zinc supplementation, zinc fortification, zinc nutrient reference values (NRVs)

## Abstract

Safe upper levels (UL) of zinc intake for children were established based on either (1) limited data from just one study among children or (2) extrapolations from studies in adults. Resulting ULs are less than amounts of zinc consumed by children in many studies that reported benefits of zinc interventions, and usual dietary zinc intakes often exceed the UL, with no apparent adverse effects. Therefore, existing ULs may be too low. We conducted a systematic bibliographic review of studies among preadolescent children, in which (1) additional zinc was provided vs. no additional zinc provided, and (2) the effect of zinc on serum or plasma copper, ceruloplasmin, ferritin, transferrin receptor, lipids, or hemoglobin or erythrocyte super-oxide dismutase were assessed. We extracted data from 44 relevant studies with 141 comparisons. Meta-analyses found no significant overall effect of providing additional zinc, except for a significant negative effect on ferritin (*p* = 0.001), albeit not consistent in relation to the zinc dose. Interpretation is complicated by the significant heterogeneity of results and uncertainties regarding the physiological and clinical significance of outcomes. Current zinc ULs should be reassessed and potentially revised using data now available for preadolescent children and considering challenges regarding interpretation of results.

## 1. Introduction

Several expert committees have proposed safe upper levels of intake (ULs) for selected micronutrients to guide consumers, clinicians, scientists, policy makers and food producers [1,2,3,4]. The UL is the level of intake considered to be safe for nearly all individuals in specific age and sex groups, when consumed for extended periods of time. The UL is established by reviewing information on the lowest level at which evidence of excessive intake occurs: the Lowest Observed Adverse Effects Level (LOAEL). When there is insufficient information to establish a LOAEL, the UL is based on the highest level at which no adverse effects are observed: the No Observed Adverse Effects Level (NOAEL). The NOAEL and possibly the LOAEL are then adjusted by dividing by an uncertainty factor (UF), generally between 1 and 10, depending on the amount and quality of available evidence and the severity and reversibility of the observed or potentially adverse effects [5,6].

Ideally, the risk assessment to establish a UL would be based on data gleaned from a number of studies, in which (1) the cause of the adverse effect can be established and the health impact measured, (2) the doses of the nutrient consumed by the study population cross the threshold at which adverse effects are first observed (LOAEL), (3) the dose is consumed consistently during a sufficient period of time for adverse effects to be manifested, and (4) data are available from various population sub-groups—defined in terms of age, sex, and physiological status [2]. When establishing the 2001 Institute of Medicine (IOM) ULs for zinc, the UL for adults was based on a LOAEL, while that for children was based on a NOAEL. The primary biomarkers that were used to identify adverse effects of excessive zinc intake were biomarkers of copper status, namely plasma concentrations of copper (Cu) or ceruloplasmin (Cp) and erythrocyte superoxide dismutase (ESOD) activity [2]. Other biomarkers of potentially adverse effects of high zinc intakes considered in establishing zinc ULs include hemoglobin (Hb) [7,8], high-density lipoproteins (HDL) or other lipids [8,9,10,11,12], iron absorption [13] or markers of iron status [8], such as ferritin (Ferr) and transferrin receptor (TfR), and magnesium balance [14]. In addition, several physiological factors must be considered when interpreting studies on the potential adverse effects of high zinc intakes. For example, the absorption of zinc from fortified food is less than from zinc supplements and varies by dose and matrix [15,16,17,18], so zinc bioavailability should be considered when establishing the UL, as has been done when estimating Recommended Intakes (RI) [1,2,3,4]. Further, several of the biomarkers for potential adverse effects of zinc, including plasma Cu, Cp, and Ferr, are affected by inflammation, so acute phase response protein (APP) markers of inflammation are needed to interpret results from these studies [19].

The World Health Organization (WHO) and the United Nations Food and Agricultural Organization (FAO), the United States Institute of Medicine (IOM), the European Food Safety Authority (EFSA), and the International Zinc Nutrition Consultative Group (IZiNCG) have published nutrient reference values for zinc, including ULs (Table 1). All five organizations derived estimates of the UL for adults based on the LOAEL, using studies that found negative effects of high-dose zinc supplements on copper status indicators [1,2,3,4]. Due to the limited data then available among young children, these organizations developed ULs using either very limited data from studies among children or extrapolations from studies of adults. For example, the IOM 2001 ULs for children were based on a single study by Walravens and Hambidge [20] that found no adverse effect of higher zinc intake on plasma Cu [2]. The International Zinc Nutrition Consultative Group (IZiNCG) later developed similar estimates for children’s ULs, using data from one study by Lind et al. [21,22], and included adjustments for bioavailability (Table 1, IZiNCG). By contrast with the IOM and IZiNCG, the WHO and ESFA committees estimated ULs for children by extrapolating data from adults to children based on metabolic rate or relative surface area [3,4], resulting in higher estimates of the ULs (Table 1).

Notably, the IOM and IZiNCG ULs are lower than the previous IOM RIs of 5 and 10 mg/day for children <12 months and 1–10 years, respectively; these ULs are only about twice the current IOM and IZiNCG RIs. This narrow range of acceptable intakes between the RI and UL severely limits dosing options for public health programs, such as large-scale zinc fortification and zinc supplementation programs.

Because the ULs derived by the IOM and IZiNCG are estimated based on a NOAEL, it is likely that children could consume more than this amount without experiencing adverse effects. Analyses of individual dietary zinc intakes by children 6–59 months of age in the US, as reported for the periods 1994–1996, 1998 [23], and 2016 [24], found that more than half of the children were consuming more zinc than the IOM ULs. Although no information was available on whether any adverse effects were associated with these higher levels of intake, there are no reports of population-wide abnormalities in copper status during this period of time.

Additional studies are now available to allow for a reassessment of current estimates of the ULs for zinc among young children, as reviewed herein. The objectives of this review are to identify available studies conducted among children who received varied levels of additional zinc intake through supplementation or zinc-fortified foods, and to determine whether there is sufficient new evidence to justify reconsidering the zinc ULs among children.

## 2. Materials and Methods

We conducted systematic bibliographic searches in PubMed to identify studies among preadolescent children in which additional zinc was provided as a supplement or fortificant vs. a no-added-zinc comparison group. Searches included the words “child + zinc” or “infant + zinc”. We also reviewed potentially relevant documents cited in these papers. Each identified abstract and selected paper was reviewed by one of two authors (D.L.d.R. and S.W.), with extracted data checked by the other of these two authors and verified by a third author (D.H.). All abstracts were first examined to identify studies that included children, reported on zinc interventions, and included a comparison group. Full papers for the potentially acceptable studies were then obtained and reviewed for the following inclusion criteria: (1) zinc supplementation or fortification was the only factor that differed between study groups, called “additional” zinc and is in addition to dietary intakes; (2) participants were randomly assigned to study group at baseline; (3) at least one biochemical marker of potential adverse effects was measured and reported either at endline or as the difference between baseline and endline (identified biomarkers to follow); (4) the intervention lasted at least one month; (5) the mean baseline age of study participants was 0–9 years even if the range of ages extended beyond this maximum over the course of the intervention period, and (6) the study population was not selected based on the presence of a specific disease state (e.g., cirrhosis or celiac disease) and the additional zinc was not provided as treatment for diarrhea or other illness for the duration of the intervention. The biomarkers that were considered as indicators of possible adverse effects are serum or plasma Cu, Cp, Ferr, TfR, and lipid concentrations; whole blood Hb, and ESOD. When full papers for potentially eligible studies were not available, we attempted to contact authors (5 cases), even though the abstracts suggested it was unlikely that the studies fit all inclusion criteria. We did not receive responses from these authors, and thus did not include these studies in this review. All data from eligible studies were extracted to Excel (Office 365, version 2012) spreadsheets and re-checked prior to entering in Review Manager 5.4 for conducting meta-analyses and creating forest plots.

### 2.1. Data Management

For consistency, we used available outcome data in the following order of priority: (1) endline mean ± standard deviation (SD) for each biomarker in the zinc intervention and comparison groups, assuming that baseline values were similar due to random allocation; (2) mean ± SD change in the biomarker concentration from baseline to endline for each group, which received lower priority because of the limited number of studies that reported the outcomes this way, and (3) conversion of other available endline data values to mean and SD of the biomarker concentration at endline (e.g., means or medians with standard error, confidence interval, interquartile ranges, and minimum/maximum ranges). For studies analyzed using option 3, we estimated the endline mean and SD biomarker concentration for the intervention and comparator groups in the following manner, as proposed in the online Cochrane Handbook for Systematic Reviews of Interventions version 6.0 [25]: (a) the formula provided by Wan et al. [26] was used when only median, 25th and 75th percentiles were presented; (b) the formula by Higgins et al. [25] was used when only 95% confidence interval (CI) and standard error were reported; (c) the equation SD = SE/√(1/NE − 1/NC) was used when only the standard error of the difference between the intervention and the control group was reported [25], (d) the calculated SD was used as an average SD for both the intervention and control group, and (e) the formula provided by Hozo et al. [27] was used when only median, minimum and maximum were reported. For comparability across studies, we transformed all reported concentrations to consistent units, as follows: serum copper was harmonized to µg/dL, ferritin to µg/L, hemoglobin to g/L; ceruloplasmin in units/L, ESOD in units/g Hb, and TfR in mg/L. In one case, only the standard error of the geometric mean was reported. We contacted authors to obtain the geometric SD, as the arithmetic SDs were used in the meta-analyses [28].

Because our meta-analyses sometimes used different comparisons than the original papers, in some cases we found a different level of significance than reported in the original publications and we present these differences in Appendix A. For example, some of the original studies used change from baseline as the primary outcome and adjusted the analyses for other covariates, such as inflammation. In the meta-analyses, we compared standardized mean difference between the intervention and the control group endline concentrations, using a random effects model without any adjustments for covariates, which is a conservative approach to identify any observable effects. This approach assumes that the randomization of study participants in all studies would balance residual confounding between treatment and control groups but does not differentiate potential effects related to changes such as reduced inflammation that may have occurred with the provision of additional zinc.

### 2.2. Statistical Analyses

We used Review Manager Software (RevMan; Version 5.4, The Nordic Cochrane Centre, The Cochrane Collaboration, 2014, Copenhagen, Denmark) for data entry and to create forest plots and related statistical analysis, such as heterogeneity. The standardized mean difference (SMD) with corresponding 95% confidence interval was estimated to evaluate the effect of the zinc intervention. SMD measures the size of the intervention effect in each study relative to the variability observed in that study. RevMan calculates heterogeneity between studies using Higgins I^2^ statistics and is considered high if I^2^ ≥ 50% [25]. We assumed heterogeneity would be high because of the diversity of characteristics in the study participants, intervention doses, chemical form of the additional zinc, intervention duration and other aspects of study design. Thus, we applied the random effect model (REM).

When a study’s interventions included more than one comparison to a single control group, such as two different zinc doses, the sample size of the control group entered into RevMan for the forest plots was divided by the number of comparison groups (generally two), to avoid applying too much weight when using the same population group as comparator.

We also conducted sensitivity analyses by separately assessing groups of studies to test for the following: mean age at baseline above or below 12 months, and whether potentially masking micronutrients were provided, such as the provision of copper to both intervention and control groups that might mask potential adverse effects of zinc on copper biomarkers, or the provision of iron that might mask an effect of zinc on ferritin or hemoglobin concentrations.

## 3. Results

### 3.1. Overall Findings

We identified 9655 articles by PubMed search, of which 222 were considered as possibly reporting on zinc intervention trials in children (Figure 1).

Following review, we ultimately identified 45 studies that fit the inclusion criteria. The primary reasons for exclusion during full paper review were because zinc was not a uniquely distinguishable component of the intervention, the publication did not provide information on possible adverse effects, duplicate reports were published from the same study, or the dose of zinc provided was uncertain.

The 44 eligible studies, with at least one biomarker of a potential adverse effect (Table 2), provided data from 21,319 children across a total of 141 group-wise comparisons (considering different doses of zinc and different outcomes). These comparisons are summarized by outcome in Appendix A.

Mean child age at baseline for individual studies ranged from 5 days to 8.5 years, with the range of ages extending up to 15 years at baseline; most available data were from infants and children < 5 years of age. The duration of the intervention period ranged from 1.5 to 18 months, and the average daily dose of additional zinc provided in these studies, either through supplementation or fortification, ranged from 0.9 to 21.4 mg/d. The maximum daily dose of additional zinc that was provided to infants was 20 mg zinc/d for 11 months, and the maximum additional daily amounts provided to children ≥ 12 months of age were 20 mg zinc/d for 12 months and 21.4 mg zinc/d for 6 months. One study provided 70 mg of supplemental zinc twice weekly, for an average daily dose of 20 mg zinc/d for up to 11 months duration in infants and up to 15 months in older children. Although dietary zinc intakes, biomarkers of inflammation, and adherence to the intervention are of interest, few studies reported this information.

Most data on potential adverse effects of additional zinc were available for three outcomes: Cu, Ferr, and Hb; therefore, this review primarily focuses on these outcomes. Limited data were also available for other outcomes and are shared in the Appendix A.

### 3.2. Copper-Related Outcomes

#### 3.2.1. Serum or Plasma Copper Outcomes

We found a total of 17 studies (*n* = 3890 children) that presented data on Cu (Appendix A) [20,21,29,30,31,32,33,34,35,36,37,38,39,40,41,42,43]. These studies provided 23 comparisons of different amounts of additional zinc, ranging from 2.3 to 21.4 mg/d, versus no additional zinc. Four of these comparisons showed a significant negative effect of additional zinc on final mean Cu concentrations [31,33,35,36]. These four comparisons included children whose initial mean ages ranged from 4 to 96 months (~8 years) who received from 3.3 to 20 mg/d additional zinc, and final mean copper concentrations in the groups that received zinc ranged from 86 to ~160 µg/dL (normal range 63–160 µg/dL [2]. By contrast, one comparison, which included infants with mean initial age of 5 days who received 4 mg/d additional zinc, showed a significant positive effect of additional zinc on Cu concentration [20]. The remaining comparisons assessed children with mean ages ranging from 5 months to ~8 years who received 2.3 to 21.4 mg/d additional zinc. Figure 2 shows a forest plot of the 23 comparisons, ordered by the amount of additional zinc. There was no significant overall effect of additional zinc on final Cu concentration (SMD = −0.11; CI: −0.27, 0.06; *p* = 0.21), and there was significant heterogeneity of results (I^2^ = 81%), which could not be explained by the amount of additional zinc provided or the children’s mean initial ages. Sensitivity analyses that removed the one study that provided additional copper to both comparison groups did not change the significance of the outcome, concerning the effect of zinc on Cu concentrations.

#### 3.2.2. Serum or Plasma Ceruloplasmin Outcomes

Three studies (*n* = 179 children), with a total of five comparisons, provided data on mean Cp concentration (Appendix A) [30,43,44]. Four of these comparisons found no significant effect of additional amounts of zinc, ranging from 5 to 15 mg/d. One study [44] found a lower final mean Cp concentration (as determined by RevMan, although not reported as significant in the original publication) among children who received 10 mg/d additional zinc in a fortified beverage (Appendix A, individual SMD = −3.83; CI: −4.60, −2.25). In this study, inflammation was not reported, and the final mean Cp concentration was 27.6 mg/dL in the zinc intervention group, which is considered to be within the normal range [2,45]. Overall, there was no significant effect of additional zinc on Cp (SMD = −1.11; CI: −2.43, 0.20; *p* = 0.10; I^2^ = 88%).

#### 3.2.3. Erythrocyte Superoxide Dismutase Outcomes (ESOD)

Four studies (*n* = 274 children), with a total of six comparisons, provided data on ESOD. All four studies found no significant effect of additional amounts of zinc, ranging from 5–20 mg/d, on ESOD concentration (Appendix A: SMD = 0.08; CI: −0.25, 0.42; I^2^ = 24%; *p* = 0.62) [30,36,43,46].

### 3.3. Iron-Related Outcomes

#### 3.3.1. Serum or Plasma Ferritin Outcomes

We found a total of 25 studies (*n* = 6649 children) that presented data on Ferr concentration (Appendix A) [21,28,29,33,34,36,38,39,43,44,46,47,48,49,50,51,52,53,54,55,56,57,58,59,60] (all data from Becquey, et al. and placebo ferritin data from Lind, et al. obtained by personal contact with authors). These studies provided a total of 39 comparisons, 20 of which compared additional zinc versus no additional zinc, and another 19 of which compared additional zinc plus iron versus additional iron. The amounts of additional zinc ranged from 0.9 to 21.4 mg/d, and the children’s mean initial ages ranged from 5.9 weeks to 8.5 years. Overall, and in both subsets of comparisons disaggregated by whether iron was also provided, there was a significant negative effect of additional zinc on final Ferr concentration (Figure 3, *p* = 0.001 overall; (a) *p* = 0.02 for comparisons with no added iron; (b) *p* = 0.006 for comparisons with added iron).

Among the comparisons of zinc versus placebo, the range of reported mean final Ferr concentrations in the groups that received zinc was 9.4 to 51.7 µg/L, with just one study reporting a final mean value that was less than 12 µg/L, the cutoff that is used to define iron deficiency among children less than five years of age. When the final mean Ferr concentrations were reported, they were 9.0 to 60.1 µg/L among the studies that included iron in both study groups, with one study reporting a final mean value < 12 µg/L. The standardized mean difference in final Ferr was considerably greater among the comparisons where additional iron was provided versus those where no iron was provided (Figure 3b: SMD = −0.63; CI: −1.07, −0.18; *p* = 0.006 vs. Figure 3a: SMD = −0.14; CI: −0.27, −0.02; *p*= 0.02), but the reported final Ferr concentrations were also much greater than observed in the comparisons that did not include iron, as demonstrated in Table 3, showing those studies with factorial design, comparing groups that received zinc vs. no zinc, and zinc plus iron vs. only iron. There was significant heterogeneity of results overall (I^2^ = 95%), but no obvious dose–response relationship. We also disaggregated studies with or without additional iron, according to a mean participant age at enrollment of less than or greater than 12 months.

When the comparisons that did not include iron were disaggregated by age, there was a significant negative effect of additional zinc among those with mean initial age ≥12 months (*p* = 0.03), but not among those <12 months (*p* = 0.13, Appendix A). In contrast, when the comparisons that did include iron were disaggregated by age, there was a significant negative effect of additional zinc among those with mean initial age < 12 months (*p* = 0.04), but not among those with children ≥ 12 months (*p* = 0.12, Appendix A). Four of the comparisons assessed the effects of added zinc and iron provided in fortified foods [33,44,54], with additional amounts of zinc ranging from 2.1 to 6.4 mg/d. None of these comparisons found a negative effect of zinc on final Ferr concentration.

#### 3.3.2. Whole Blood Hemoglobin Outcomes

We found a total of 35 studies (*n* = 18,989 children) that presented data on whole blood hemoglobin (Hb) concentration (Appendix A) [21,28,29,32,33,34,36,37,38,39,40,43,44,46,48,49,50,51,52,53,54,55,56,57,58,59,60,61,62,63,64,65,66,67,68,69]. These studies provided a total of 56 comparisons, 33 that compared additional zinc versus no additional zinc and 23 that compared additional zinc plus iron versus additional iron. The amounts of additional zinc ranged from 1.2 to 21.4 mg/d, and the children’s mean initial ages ranged from 5.9 weeks to 8.5 years. There was no significant effect of zinc on final Hb concentration, regardless of whether iron was also provided (Figure 4, overall SMD: −0.02; CI: −0.08, 0.03), and there was no evidence of a dose–response relationship.

There was significant heterogeneity in the results (I^2^ = 59%), though less than with the copper and ferritin results. Again, there was no clear pattern of results according to the initial ages of the children or doses of zinc. Among the seven comparisons that found a negative effect from zinc, the mean final Hb concentrations ranged from 102 to 115 g/L in those who received zinc compared to 106 to 119 g/L in those who did not receive zinc [21,52,58,59,60,64,65]. When we only included comparisons in which no iron was provided as a supplement or fortificant to both study groups, there was no significant effect of zinc on Hb (SMD: 0.01; CI: −0.06, 0.08), I^2^ = 54%) (Figure 4a). As with ferritin, we also disaggregated those studies in which additional iron was provided, (Figure 4b) and found no significant effect of zinc (SMD: −0.07; CI: −0.17, 0.03), I^2^ = 66%). Likewise, when the studies were disaggregated by mean initial age, there were no significant effects of zinc in either age group (Appendix A, <12 months: SMD: −0.07; CI: −0.14, 0.01, *p* = 0.071; ≥12 months SMD: 0.04; CI: −0.05, 0.14, *p* = 0.35).

#### 3.3.3. Serum Transferrin Receptors

Four studies (*n* = 1518 children), with a total of eight comparisons, provided data on TfR. All four studies found a significant positive effect of additional amounts of zinc, ranging from 0.9–10 mg/day on TfR concentration. (Appendix A: SMD: 0.22; CI: 0.01, 0.43, *p* = 0.04) [21,29,49,51].

### 3.4. Lipid Outcomes

We found a total of four studies (*n* = 398 children; four comparisons) that presented data on some marker of lipid metabolism (Appendix A) [20,35,43,70]. The amounts of additional zinc ranged from 2.8 to 10 mg/d, and the children’s mean initial ages ranged from 5 days to 8 years. One of these comparisons [70] showed a significant effect from additional zinc on mean nervonic acid concentration. This comparison was conducted in children with mean initial ages of approximately 8 years, with a dose of 2.8 mg/d additional zinc. The remaining three comparisons assessed children with mean initial ages ranging from 5 days to 5 years of age who received 3.75 to 10 mg/d additional zinc and reported no difference in serum cholesterol concentration with additional zinc. Because there were so few results, the actual biomarker differed across studies, and the implication of a positive or negative change in concentration was not clear for those markers available; we did not conduct forest plot analyses of these studies.

## 4. Discussion

Our literature search identified a sizeable number of relevant studies that were not available or considered when international authorities developed the existing zinc ULs for infants and children. Most of the comparisons on the effects of zinc on potential adverse outcomes that were identified in the present review provided doses of additional zinc that are greater than the current EFSA, IOM and, IZiNCG ULs. Thus, considerably more information is now available to reassess the ULs for infants and children, and it may no longer be necessary to extrapolate from the results of studies in adults.

Overall, there were no significant effects of providing as much as 21 mg zinc daily (in addition to zinc intake from the diet) for extended periods of time to infants and children on markers of Cu status or Hb concentration. These results for Hb concentration are consistent with an earlier meta-analysis by Dekker and Villamor [7]. Although a few individual studies found lower Cu status markers or Hb concentrations among children who received additional zinc, other studies found the opposite outcome and there was no observable relationship between the dose of zinc provided or the ages of the study participants and these outcomes. The heterogeneity of results could not be explained with the information examined in the current analyses. Thus, it seems that there is no clear LOAEL within the available dosage range based on these biomarkers of potential adverse effects of zinc.

By contrast with the Cu and Hb outcomes, providing additional zinc resulted in significantly lower Ferr; this was more pronounced in cases where additional iron was also given to both study groups. However, the final mean ferritin concentrations were greater than the cut-off for iron deficiency among children less than 5 years of age (12 µg/L [71]), in almost all comparison groups that received additional zinc. In other words, it appears that additional zinc may have reduced the uptake of supplemental iron, but it did not cause iron deficiency in most children who received zinc. Further, despite the negative effect of zinc, in the groups that received iron in addition to zinc, the final Ferr was still substantially greater than when no iron was given (Table 3), so it seems that any negative effect of zinc on iron status can be mitigated by providing additional iron. Thus, it is uncertain whether this outcome on Ferr should be considered an adverse effect of zinc when developing the UL. Finally, there was no consistent pattern of effect by age group, and there was no significant effect of providing up to 6.4 mg additional zinc/d in fortified foods.

As noted above, during these analyses, we discovered several dilemmas when assessing the possible adverse effects of zinc. Among the main adverse outcomes considered, Cu, Cp, and Ferr are all acute phase proteins that respond markedly to systemic inflammation [19]. Thus, if providing additional zinc reduced the risk of infection and inflammation, these biomarkers could have declined in response to the reduced inflammation rather than due to an adverse effect of zinc on copper or iron uptake or status. Regrettably, many available studies did not report information on the participants’ inflammation status; therefore, our analyses did not adjust for this possible effect.

A second concern regarding the interpretation of results of individual studies relates to the final values of the markers of possible adverse effects, and whether any of the negative effects that were observed in a few of the studies are of clinical significance. In those studies that reported lower final Cu in the zinc groups, for example, the mean final Cu concentrations were all within the normal range [2], so it is uncertain whether these differences represent a clinically important adverse effect of zinc. A similar issue regarding Ferr is described above.

A third issue relates to the lack of a clear dose–response relationship or response threshold in any of the sets of comparisons, which makes it difficult to identify a definitive LOAEL. There is currently no clear guidance on how to interpret such inconsistent results when establishing a UL.

A fourth challenge is the need to adjust zinc intake for bioavailability, as is done for the dietary requirements. Most of the studies identified for this review provided additional zinc in the form of a supplement. There is evidence to indicate that the effect of zinc intake on zinc absorption, and possibly on uptake or metabolism of other minerals, differs when zinc is provided as a supplement between meals (or to fasting individuals), rather than with meals. For example, Sandstrom et al. [72] found 58% zinc absorption from an aqueous solution containing 40 µmol zinc and 40 µmol iron following an overnight fast, whereas zinc absorption fell to 25% when the same solution was provided with a meal composed of rice and meat sauce. Sian et al. [73] measured zinc absorption in four adults who consumed zinc supplements with or without meals. Zinc absorption was considerably lower in all subjects when zinc was consumed with meals compared with fasting, regardless of the phytate content in the meal. Moreover, several studies indicate that the effect of providing additional zinc on plasma zinc concentration is significantly greater when administered as a supplement compared with zinc-fortified foods. For example, in a 6-month longitudinal study comparing the effects of the same amount of zinc delivered in a liquid supplement or in a fortified food, the final plasma zinc concentration was significantly greater among those who received the supplement [33]. Likewise, two short-term studies in Senegal found that plasma zinc concentration responded to supplementation, but not fortification, both in children [74] and adults [75]. These studies consistently demonstrate that zinc absorption from supplements is greater when administered alone and apart from meals compared with absorption from the same amount of additional zinc when provided with meals, so any adverse effects of zinc may also be greater when the zinc is delivered as supplements apart from meals. Regrettably, most trial reports do not state exactly how the supplemental zinc doses were delivered, which complicates the interpretation of these results.

Finally, the results reported herein only reflect the additional zinc provided via supplementation or fortification during the study’s intervention period and do not include zinc that is contributed by the participants’ usual dietary intake. Use of only the reported doses of additional zinc to establish a UL would underestimate the total zinc intake from both the diet and the additional zinc delivered as part of the study.

Several weaknesses in the current analyses must be recognized. Firstly, we did not have sufficient resources to complete a formal systematic review, and we did not conduct a risk of bias assessment nor evaluate individual study quality. Secondly, we were not able to communicate systematically with the authors of all the studies identified to solicit additional information or clarifications, although we did obtain inputs in some cases where the study methods or results reported in the published papers were unclear. Despite these limitations, we were able to achieve our primary objective of determining whether sufficient new information is available to merit reconsideration of the UL in children. Our conclusion is that there is indeed sufficient new evidence to reassess these ULs, and we believe that this effort will be valuable both because of the limited information available previously and the fact that the current ULs seem lower than necessary. Not only do substantial proportions of several higher-income populations habitually consume more than the current ULs without apparent detrimental health effects [23,24], but the narrow ranges between the RIs and the ULs complicate efforts to develop practical dietary guidance or prepare feasible intervention programs in situations where there is a high risk of zinc deficiency. For these reasons, we strongly encourage global health and nutrition authorities to reconsider the ULs based on the newly available information.

Our efforts to conduct meta-analyses using data from multiple studies highlight issues for consideration by those wishing to facilitate the use of their data for future meta-analyses. We recommend that authors consider presenting the following in their paper or Appendix A: (1) data on inflammation and related adjustments, (2) raw means and SDs, in addition to any adjusted or transformed values, (3) dietary intakes in addition to any additional dose provided, (4) change from baseline as well as final values, (5) actual sample sizes by indicator, (6) population proportions outside normal ranges at baseline and endline, (7) data split across age ranges that match RI/UL age groups, (8) differences in outcomes by initial status (low/normal/high), and (9) how to access datasets for meta or pooled analyses.

Because of the concerns that surfaced regarding the interpretation of available studies, new deliberations on the UL, ideally, should involve the investigators of the original studies, who could possibly provide access to individual participant data and information on the participants’ dietary zinc intake and markers of inflammation, where available. Further, consensus must be developed on how best to interpret individual studies that reported significant effects of zinc, even when the meta-analyses found no significant effects overall. We believe that these decisions should be made by duly constituted authoritative bodies to ensure broad global acceptance and harmonization of recommendations. In conclusion, a considerable body of new evidence is available to permit reassessment of the ULs for zinc among infants and young children, and we encourage the relevant international authorities to proceed with this task.

## Figures and Tables

**Figure 1 nutrients-14-01938-f001:**
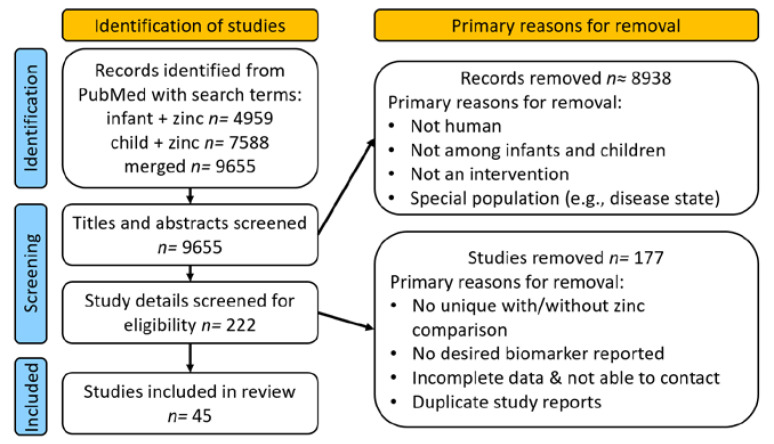
Flow of study identification, screening, and inclusion.

**Figure 2 nutrients-14-01938-f002:**
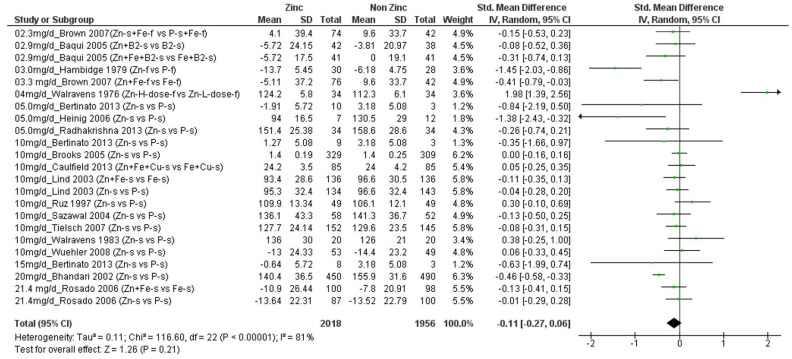
Effect of additional zinc intervention on plasma or serum copper concentrations among children. Figure 2 legend: The amount of additional zinc provided in the zinc groups are shown, along with the first author and year of publication. The intervention groups are indicated, as follows. Zn-s: Zinc supplement, Zn+ Fe-s: Zinc Plus Iron supplement, Zn + Fe + Cu-s: Zinc plus iron plus copper supplement, Fe + Cu-s: Iron plus copper supplement, Zn + B2-s: Zinc plus vitamin B12 supplement, B2-s: vitamin B2 supplement, Zn + Fe + B2: zinc plus iron plus vitamin B2 supplement; Zn-f: zinc fortified, P-s: placebo supplement, P-f: Placebo fortified, Fe-f: iron fortified, Fe-s: iron supplement, Zn-H-Dose-f: Zinc high dose fortified, Zn-L-Dose-f: zinc low dose fortified. Green dots represent the means for each plot.

**Figure 3 nutrients-14-01938-f003:**
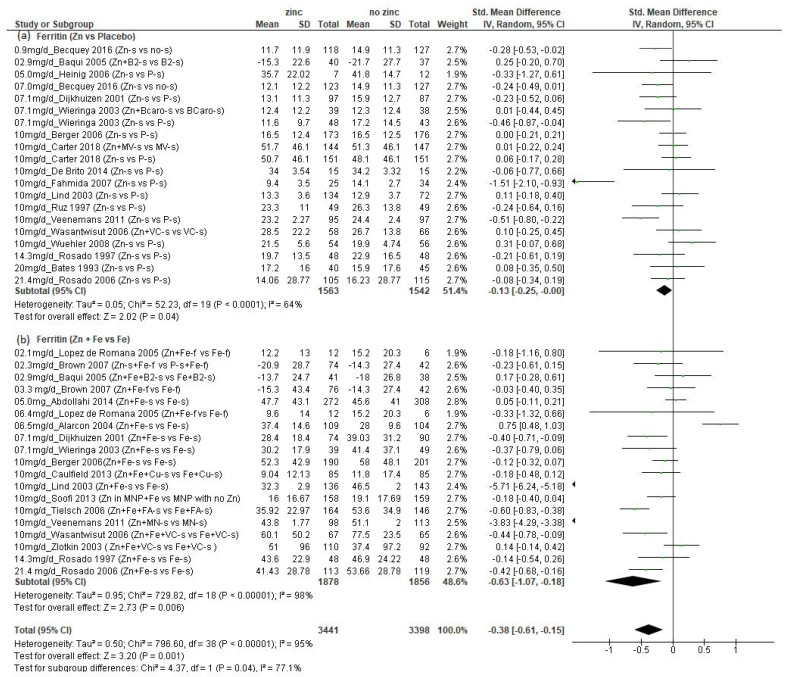
Effect of additional zinc intervention on plasma or serum ferritin concentrations among children: (**a**) with no additional iron, (**b**) with additional iron. Figure 3 legend: The amount of additional zinc provided in the zinc groups are shown, along with the first author and year of publication. The intervention groups are indicated, as follows. Zn-s: Zinc supplement; Zn+ Fe-s: Zinc plus iron supplement; Zn + Fe + Cu-s: Zinc plus iron plus copper supplement; Fe + Cu-s: Iron plus copper supplement; Zn + B2-s: Zinc plus vitamin B12 supplement; B2-s: Vitamin B2 supplement; Zn + Fe + B2-s: Zinc plus iron plus vitamin B2 supplement; P-s: Placebo supplement; Fe-f: Iron fortified; Fe-s: Iron supplement; no-s: No supplement; Zn + BCaro-s: zinc plus beta carotene supplement; BCaro-s: Beta carotene supplement; Zn + MV-s: Zinc plus multivitamin supplement; MV-s: Multivitamin supplement; Zn + VC-s: Zinc plus vitamin C supplement; VC-s: Vitamin C supplement; Zn + Fe-f: Zinc plus iron fortified; Fe-f: Iron fortified; Zn-s + Fe-f: Zinc supplement plus iron fortified; P-s + Fe-f: Placebo supplement plus iron fortified; Zn + Fe + FA-s: Zinc plus iron plus folic acid supplement; Fe + FA-s: Iron plus folic acid supplement; Zn in MNP: Micronutrient powder with zinc plus iron; MNP with no Zn: Micronutrient powder with no zinc plus iron; Zn + MN: Zinc plus micronutrient supplement; Zn + Fe + VC-s: Zinc plus iron plus vitamin C supplement. NOTE: placebo (P) concentration reported in Lind 2003 publication updated by Dr. Lind (personal communication).

**Figure 4 nutrients-14-01938-f004:**
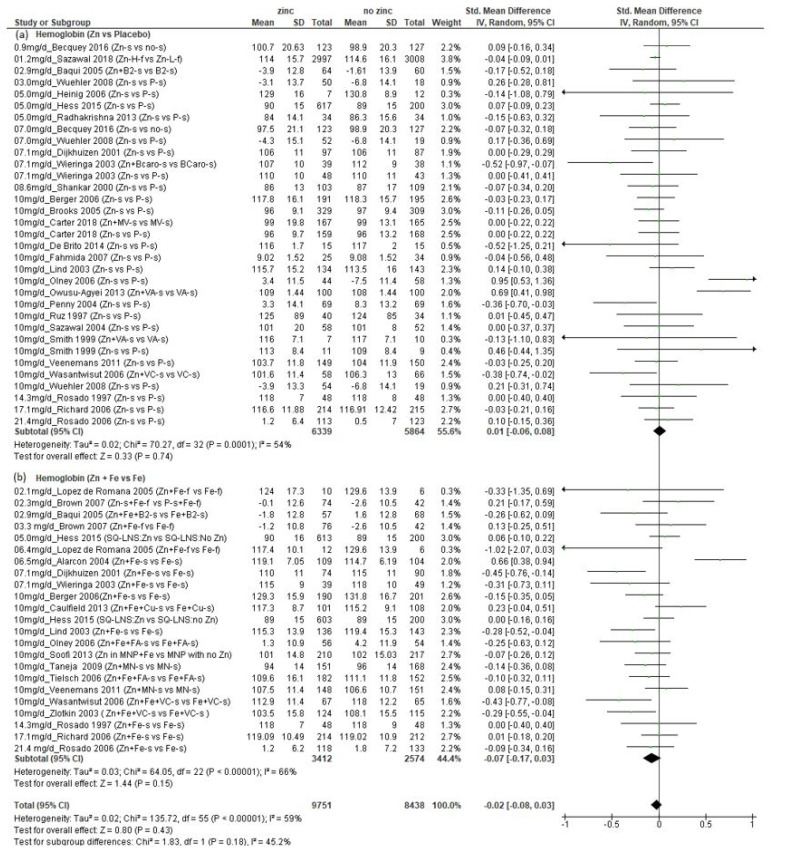
Effect of additional zinc intervention on whole blood hemoglobin concentrations: (**a**) with no additional iron, (**b**) with additional iron. Figure 4 legend: The amount of additional zinc provided in the zinc groups are shown, along with the first author and year of publication. The intervention groups are indicated, as follows. Zn-s: Zinc supplement; Zn+ Fe-s: Zinc plus iron supplement; Zn + Fe + Cu-s: Zinc plus iron plus copper supplement; Fe + Cu-s: Iron plus copper supplement; Zn + B2-s: Zinc plus vitamin B12 supplement; B2-s: Vitamin B2 supplement; Zn + Fe + B2-s: Zinc plus iron plus vitamin B2 supplement; Zn-f: Zinc fortified; P-s: Placebo supplement; P-f: Placebo fortified; Fe-f: Iron fortified; Fe-s: Iron supplement; Zn-H-Dose-f: Zinc high dose fortified; Zn-L-Dose-f: Zinc low dose fortified; Zn-H-f: Zinc high fortified; Zn-L-f: Zinc low fortified; no-s: No supplement; Zn + BCaro-s: zinc plus beta carotene supplement; BCaro-s: Beta carotene supplement; Zn + MV-s: Zinc plus multivitamin supplement; MV-s: Multivitamin supplement; Zn + VA-s: Zinc plus vitamin A supplement; VA-s: Vitamin A supplement; Zn + VC-s: Zinc plus vitamin C supplement; VC-s: Vitamin C supplement; Zn + Fe-f: Zinc plus iron fortified; Fe-f: Iron fortified; Zn-s + Fe-f: Zinc supplement plus iron fortified; P-s + Fe-f: Placebo supplement plus iron fortified; SQ-LNS:Zn: Small quantity Lipid Nutrient supplement with zinc; SQ-LNS: No Zn: Small quantity Lipid Nutrient supplement with no zinc; Zn + Fe + FA-s: Zinc plus iron plus folic acid supplement; Fe + FA-s: Iron plus folic acid supplement; Zn in MNP: Micronutrient powder with zinc plus iron; MNP with no Zn: Micronutrient powder with no zinc plus iron; Zn + MN: Zinc plus micronutrient supplement; Zn + Fe + VC-s: Zinc plus iron plus vitamin C supplement.

**Table 1 nutrients-14-01938-t001:** Current estimates of daily dietary zinc requirements (EAR or NR, RDA or RNI) and estimated safe upper levels of zinc intake (UL), in milligrams/day, as proposed by IOM, IZiNCG and FAO/WHO for children and adolescents, and doses of supplemental zinc provided in reviewed studies ^1^.

IOM [2]				IZiNCG [1]			EFSA [4]		WHO/FAO [3]				
Age Range	EAR †	RDA *	NOAEL UL ‡	EAR † ^mix/unref	RDA * ^mix/unref	NOAEL UL ‡	Age Range	NOAEL UL ‡‡	Age Range	Ref wt ^2^ kg	NR ^^h/m/L Avail.	RNI ^^h/m/L Avail.	LOAEL UL ‡‡
0–6 m	-	2	4	-	-	-	-	-	0–3 m	-	-	1.1/2.8/6.6	-
									3–6 m	6	0.5/1.2/2.9	-	
									6–12 m	-	1.7/2.8/5.6	-	
7–11 m	2.5	3	5	3/4	4/5	6	-	-	7–12 m	9	-	2.5/4.1/8.4	13
1–3 y				2/2	3/3	8	1–3 y	7	1–3 y	12	1.7/2.8/5.5	2.4/4.1/8.3	23
3–6 y				3/4	4/5	14			3–6 y	17	1.9/3.2/6.5		23
							4–6 y	10	4–6 y	-		2.9/4.8/9.6	
4–8 y	4.0	8	12										
							7–10 y	10	6–10 y	25	2.3/3.7/7.5		28
									7–9 y			3.3/5.6/11.2	
9–13 y	7.0	8	23	5/7	6/9	26	11–14 y	18					
									Male/Female				
									10–18 y F			4.3/7.2/14.4	
									10–18 y M			5.1/8.6/17.1	
									10–12 y F	47	3.2/5.3/10.7		32
									10–12 y M	49	3.9/6.5/13.1		34
									12–15 y F	47	3.0/5.0/10.1		36
									12–15 y M	49	3.7/6.2/12.4		40
14–18 y F	7.3	9	34	7/9	9/11	39	15–17 y	22	15–18 y F	47	2.6/4.4/8.8		38
14–18 y M	8.5	11	34	8/11	10/14	44			15–18 y M	49	3.0/5.0/10.0		48

^1^ Original to this manuscript using data from references cited. IOM = United States Institute of Medicine, IZiNCG = International Zinc Nutrition Consultative Group, WHO/FAO = the World Health Organization/Food and Agriculture Organization of the United Nations. † EAR = estimated average requirement; similar to NR = normative requirements, as measures of physiological requirement, NRs converted to mg/day using reported reference weights from reference: [3]. * RDA = recommended dietary allowance; similar to RNI = recommended nutrient intake, as estimates of dietary intake required to meet the physiological requirements of most (>97%) individuals, and similar to currently recommended RI = Recommended Intake. ‡ UL estimated based on No Observed Adverse Effects Level. ‡‡ UL estimated based on Lowest Observed Adverse Effects Level in adults. ^2^ ref wt kg = reference weight in kilograms. ^mix/unref = mixed/unrefined as follows: Mixed: refined vegetarian or mixed diets, such as those with phytate:zinc molar ratios ≤ 18. Unrefined: unrefined cereal-based diets, such as those with phytate:zinc molar ratios > 18. ^^h/m/l avail. = high/medium and low availability as follows: High availability: Refined diets low in cereal fiber, low in phytic acid content, and with phytate–zinc molar ratio < 5; adequate protein content principally from non-vegetable sources, such as meats and fish. Includes semi-synthetic formula diets based on animal protein. Moderate availability: Mixed diets containing animal or fish protein. Lacto-ovo, ovo-vegetarian, or vegan diets not based primarily on unrefined cereal grains or high-extraction-rate flours. Phytate–zinc molar ratio of total diet within the range 5–15, or not exceeding 10 if more than 50% of the energy intake is accounted for by unfermented, unrefined cereal grains and flours and the diet is fortified with inorganic calcium salts (>1 g Ca^2+^/day). Availability of zinc improves when the diet includes animal protein or milks, or other protein sources or milks. Low availability: Diets high in unrefined, unfermented, and ungerminated cereal grain, especially when fortified with inorganic calcium salts and when intake of animal protein is negligible. Phytate–zinc molar ratio of total diet exceeds 15, high-phytate, soya-protein products constitute the primary protein source. Diets in which, singly or collectively, approximately 50% of the energy intake is accounted for by the following high-phytate foods: high-extraction-rate (≥90%) wheat, rice, maize, grains and flours, oatmeal, and millet; chapatti flours and tanok, and sorghum, cowpeas, pigeon peas, grams, kidney beans, black-eyed beans, and groundnut flours. High intakes of inorganic calcium salts (>1 g Ca^2+^/day), either as supplements or as adventitious contaminants (e.g., from calcareous geophagia), potentiate the inhibitory effects and low intakes of animal protein exacerbates these effects. m = months, y = years, M = male, F = female.

**Table 2 nutrients-14-01938-t002:** Total number of comparisons identified, and numbers identified by zinc vs. no zinc comparisons available in all identified studies by age group and outcome ^1^.

Ages ^2^	Serum or Plasma Copper	Serum Ceruloplasmin	ESOD	Serum Ferritin	Hemoglobin	Serum Transferrin Receptor	Lipids	Dose Ranges of Studies by Age ^3^ (mg)
0–5 months	1	0	0	2	1	0	1	4–5
6–12 months	11	0	1	20	25	6	1	2.3–10
1–3 years	3	1	2	11	18	2	0	0.9–20
4–6 years	5	0	0	5	9	0	1	1.2–21.4
7+ years	3	4	3	1	3	0	1	2.8–17.1
Total	23	5	6	39	56	8	4	0.9–21.4

Original to this manuscript: ^1^ Sorting is subjective, using the mean initial age and the time period in which most of the intervention took place (for example, if the mean initial age was 4 months and the study lasted 6 months, the study was counted in the 6–12-month age range); 44 total studies provided these 141 comparisons (Appendix A). ^2^ Age ranges are similar to those used to develop RI and UL recommendations (Table 1). ^3^ Daily supplemental doses of zinc provided in the studies in this review, by mean initial age (Appendix A).

**Table 3 nutrients-14-01938-t003:** Relative ferritin concentration (µg/L) at endline compared to placebo in studies with factorial design: both zinc plus iron compared to iron, and zinc compared to placebo ^1^.

Intervention Group	Zinc + Iron	Iron	Zinc	Placebo
Study:				
Baqui [29]	8.0 ^a^	3.7 ^a^	6.4 ^a^	0 ^a^
Berger [50]	35.8 ^b^	41.5 ^b^	0 ^a^	0 ^a^
Dijkhuizen [52]	12.3 ^b^	22.5 ^b^	−2.3 ^a^	0 ^a^
Lind [21] ^2^	18.4 ^b^	32.6 ^c^	−0.6 ^a^	0 ^a^
Rosado 1997 [55]	20.7	24.0	−3.2	0 ^a^
Rosado 2006 [38]	25.2 ^b^	37.4 ^b^	−2.1 ^a^	0 ^a^
Wasantwisut [58]	26.5 ^b^	45.4 ^c^	0.2 ^a^	0 ^a^
Weiringa [59]	16.5 ^b^	23.0 ^b^	−4.2 ^a^	0 ^a^

^1^ Values are mean or median endline ferritin concentration (or change in concentration from baseline) for each intervention group, less the mean endline concentration in the placebo group. ^2^ Placebo concentration reported in Lind 2003 publication updated by Dr. Lind (personal communication). ^a,b,c^ Values in a row with no similar letter in superscript are significantly different; Rosado ‘97 only reported differences from baseline. Original to this manuscript.

## Data Availability

Supporting Materials include supporting data tables on included studies and Appendix A.

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
