# Peer review of "Reconsidering the Tolerable Upper Levels of Zinc Intake among Infants and Young Children: A Systematic Review of the Available Evidence"

_nutrients, 2022, doi:10.3390/nu14091938_

Round 1

Reviewer 1 Report

This manuscript addresses an important and much needed gap in the current recommendations for zinc intakes in children. There are several specific areas that make this a strong and useful report.

  1. The authors used rigorous inclusion and exclusion criteria for the work they reviewed.  
  2. They evaluated a wide range of biomarkers in their analysis. 
  3. They transformed/standardized units to permit comparisons between studies. 
  4. The figures were detailed and helpful. 

Author Response

Thank you for your gracious review!

Reviewer 2 Report

I was so excited to read this manuscript! The importance of this research in the field cannot be highlighted enough. The manuscript is well written, and the limitations and challenges of the manuscript are transparently listed. 

A few minor comments/questions:

what would the authors suggest as a logical next step to advance this research field? 

Lines 424 to 426: The authors conclude that ," there were no significant effects of providing as much as 21 mg additional zinc daily for extended periods of time to infants and children on markers of Cu status or Hb concentration. (Additional to what?)

Author Response

Thank you for your enthusiastic review.  Responses as follows:

line 424-426: use of "additional" zinc:

line 136 in methods is updated to clarify the use of "additional"

Suggested logical next steps:  added lines 516-525 in relation to logical next steps for research, beyond what is currently recommended in the final paragraph.

Please note that we also addressed one more challenge that was missed in the previously reviewed version, lines 469-493, and incorporated minor edits throughout.